# The impact of health inequity on spatial variation of COVID-19 transmission in England

**Thomas Rawson** [ID][1]*, **Wes Hinsley**[1], **Raphael Sonabend**[1], **Elizaveta Semenova**[2], **Anne Cori**[1], **Neil M Ferguson**[1,3]

**1** MRC Centre for Global Infectious Disease Analysis, Jameel Institute, School of Public Health, Imperial College London, London, United Kingdom, **2** Department of Epidemiology and Biostatistics, Imperial College London, London, United Kingdom, **3** National Institute for Health Research Health Protection Research Unit in Modelling Methodology, Imperial College London, Public Health England, London School of Hygiene & Tropical Medicine, London, United Kingdom

* t.rawson@imperial.ac.uk

**Data Availability Statement:** The source code and data used to produce the results and analyses presented in this manuscript are available from our

## Abstract

Considerable spatial heterogeneity has been observed in COVID-19 transmission across administrative areas of England throughout the pandemic. This study investigates what drives these differences. We constructed a probabilistic case count model for 306 administrative areas of England across 95 weeks, fit using a Bayesian evidence synthesis framework. We incorporate the impact of acquired immunity, of spatial exportation of cases, and 16 spatially-varying socio-economic, socio-demographic, health, and mobility variables. Model comparison assesses the relative contributions of these respective mechanisms. We find that spatially-varying and time-varying differences in week-to-week transmission were definitively associated with differences in: time spent at home, variant-of-concern proportion, and adult social care funding. However, model comparison demonstrates that the impact of these terms is negligible compared to the role of spatial exportation between administrative areas. While these results confirm the impact of some, but not all, static measures of spatially-varying inequity in England, our work corroborates the finding that observed differences in disease transmission during the pandemic were predominantly driven by underlying epidemiological factors rather than aggregated metrics of demography and health inequity between areas. Further work is required to assess how health inequity more broadly contributes to these epidemiological factors.

## Author summary

During the COVID-19 pandemic, different geographic areas of England saw different patterns in the number of confirmed cases over time. This study investigated whether demographic differences between these areas (such as the amount of deprivation, the age and ethnicity of the populations, or differences in where people spent their time) were linked to these differences in disease transmission. We also considered whether this was associated with the number of cases in neighbouring areas as well. Using a mathematical model fit to multiple data streams, we discovered that a statistically significant link between some demographic variables (time spent at home, COVID-19 variant, and the amount of adult

GitHub repository: https://github.com/thomrawson/Rawson-spatial-covid.

**Funding:** TR acknowledges funding by Community Jameel and from the MRC Centre for Global Infectious Disease Analysis (reference MR/X020258/1), funded by the UK Medical Research Council (MRC). This UK funded award is carried out in the frame of the Global Health EDCTP3 Joint Undertaking. ES acknowledges support in part by the AI2050 program at Schmidt Futures (Grant [G-22-64476]). The funders had no role in study design, data collection and analysis, decision to publish, or preparation of the manuscript.

**Competing interests:** I have read the journal's policy and the authors of this manuscript have the following competing interests: AC has received payment from Pfizer for teaching of mathematical modelling of infectious diseases. All other authors declare no competing interests.

social care funding) and week-to-week transmission exists, but this relationship is very small, and the influence of cases in neighbouring areas was far more impactful in explaining differences in transmission between areas over time.

## Introduction

During the COVID-19 pandemic, measures of deprivation have been identified as impacting health outcomes, with more deprived areas reporting higher COVID-19 attributed mortality, both in England [1] and globally [2]. Less well-understood is the impact these measures have on disease incidence–confirmed cases. Descriptive studies early in the pandemic identified that English administrative areas with a higher Index of Multiple Deprivation (IMD) reported more cases of COVID-19 than those with lower IMD scores [3,4] during the first wave of disease incidence in 2020. However, such patterns do not persist throughout the entire epidemic, and for some periods the opposite trend can now be observed (Fig 1B).

England is divided into Lower Tier Local Authorities (LTLA)–areas of social service provisioning (see section 1.1 of S1 Appendix). For these different areas, data is available on the socio-demographic makeup–the average age, ethnic population proportions, population density; on socio-economic metrics–median earnings, employment, education; and epidemiological data throughout the pandemic–daily new cases, variant proportions, COVID-19 support funding allocated, and mobility data recording time spent at different locations. These variables vary greatly across LTLAs, and similarly disease incidence and rates of infection have varied across LTLAs during the pandemic [5]. Fig 1 demonstrates spatial variation in two covariates of interest, and mean per capita weekly incidence of COVID-19 stratified by these variables (see section 1.6 of S1 Appendix for plots of other covariates). Some variables change weekly (community mobility, variant proportion), others change annually (funding allocation, income), while others are fixed by LTLA for the entire duration.

Recognising the variation in disease incidence across administrative areas, the UK government briefly implemented a tiered lockdown system on October 14th 2020 [6], where more stringent rules on social mixing were applied to areas of the country with a greater incidence of COVID-19. This system was retired the following month for a second nationwide stay-at-home order. It remains unclear as to whether these observed epidemiological differences can be explained solely by spatial drivers of disease spread, or whether the intrinsic factors associated with each LTLA influenced the epidemic trajectory in each respective LTLA. Hypothetically, for example, populations in wealthier LTLAs may have been more able to work from home, or may have had more access to space to self-isolate in. LTLAs with a higher proportion of elderly residents may have been more susceptible to infection, or may have seen less social mixing. LTLAs that received more COVID-19 support funding per head, may have subsequently achieved better disease suppression.

Here, we model the number of weekly pillar 2 (general population testing) PCR-confirmed COVID-19 cases in 306 English LTLAs, for 95 weeks, from the week beginning May 10th 2020 to the week beginning February 27th 2022. We assume the number of weekly cases in an LTLA is determined by the previous week's number of cases, plus a proportion of imported infections from adjacent LTLAs controlled by a model parameter. Additional model parameters then control the relative influence of 16 socio-economic and -demographic variables, and time- and LTLA- varying terms on the observed increases and decreases in cases. Model parameters are fit to English COVID-19 surveillance data by LTLA through a Bayesian evidence synthesis framework.

## Variation in socio-demographic factors by LTLA and the respective stratification of per capita COVID-19 incidence

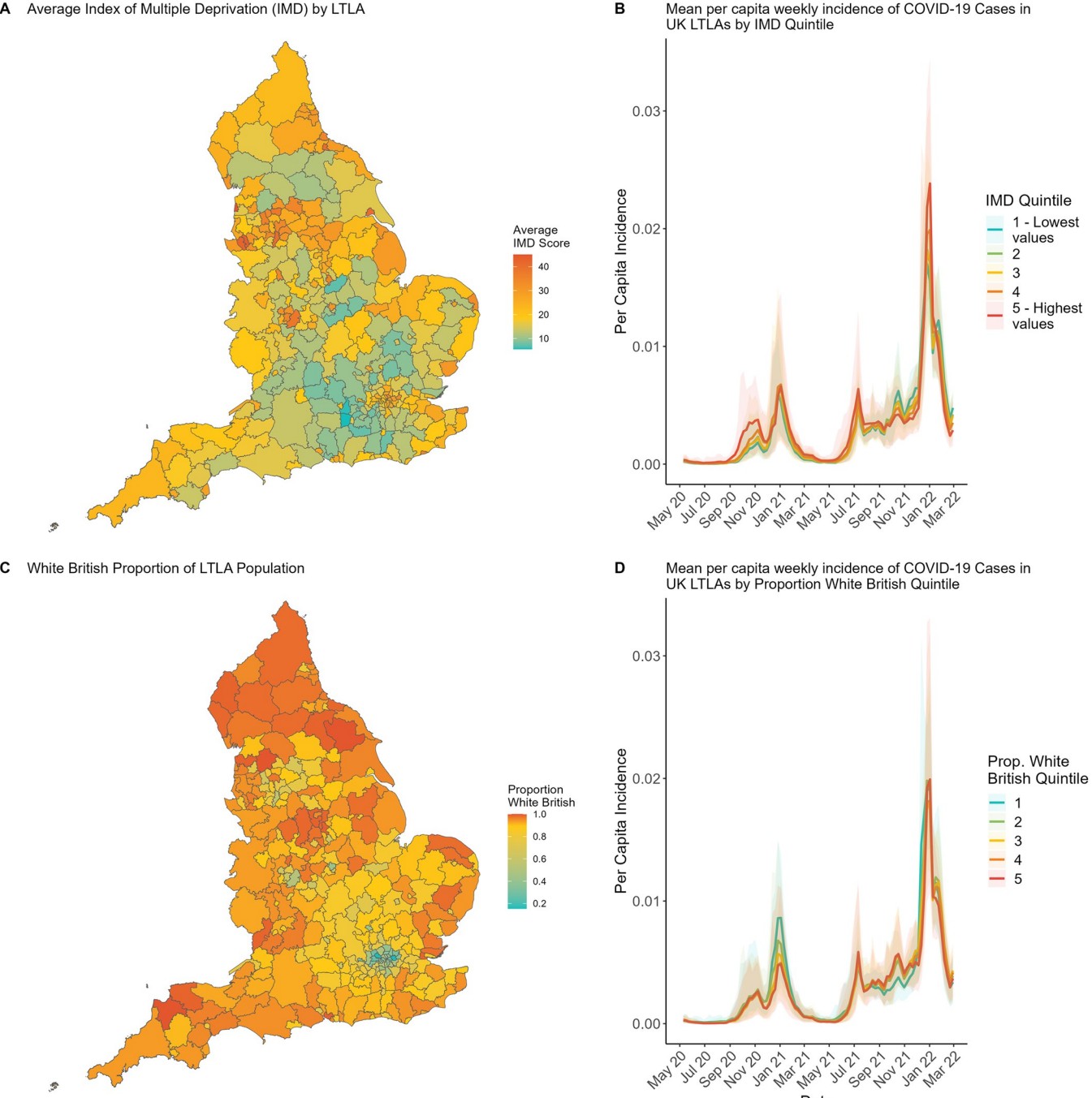

**Fig 1. Variation in socio-demographic factors by LTLA, and the respective differences in average per capita incidence of COVID-19 cases when stratified by these socio-demographic factors. (A/C)** Plots depict how IMD and White British population proportion vary across the 306 English LTLAs we consider. **(B/D)** The 306 considered LTLAs are partitioned into quintiles (blue being the lowest values quintile and red the highest values quintile) based on their (B) IMD scores and (D) White British population proportion respectively. Lines display the mean per capita weekly incidence of COVID-19 across all LTLAs in each quintile. Shaded regions depict the 95% quantiles. Quintile binning in plots B/D is for illustrative purposes—model fitting is performed to the continuous measures presented in plots A/C. Boundary source: Office for National Statistics licensed under the Open Government Licence v.3.0 [29]. Contains OS data crown copyright and database right (2024).

Real-time modelling studies provided valuable insights and projections into key epidemiological parameters throughout the pandemic [7], through regular reports integrating the latest epidemiological data. In this study we investigate how the composition of a population, and population-level covariates, contributes to week-to-week transmission potential. Identification of such contributions would inform whether real-time-modelling efforts could be improved in the future by integrating such socio-economic and -demographic data.

## Results

### Model fit

The model successfully captured the variation in LTLA-specific epidemic trajectories. Fig 2A sums the model fit across all 306 LTLAs, while plots 2B and 2C show the model fit, for example, to the two LTLAs with the greatest variation in their epidemic trajectory as assessed via dynamic time warping (DTW) distance [8] (a metric for analysing similarity in time series data). All LTLAs see broadly three principle epidemic waves, initiated by the emergence of the Alpha, Delta, and Omicron variants respectively.

The effective reproduction number, $R_{i,t}^{eff}$, is an epidemiological parameter dictating the number of secondary infections caused by a primary infection in LTLA $i$ at week $t$. Hence, when $R_{i,t}^{eff} > 1$, cases are observed to increase in LTLA $i$ at week $t$. Likewise, when $R_{i,t}^{eff} < 1$, cases are observed to decrease. Our model assumes that $R_{i,t}^{eff}$ is made up of three principle elements. First, a time-varying random walk term, $z_t$, observed across all LTLAs, capturing the impact of multiple time-varying factors such as changes to non-pharmaceutical interventions (NPIs), vaccination uptake, school closures, and national holidays. Second, an LTLA-varying error term, $\theta_i$, to capture any unexplained intrinsic differences between the reproduction number across LTLAs. Third, a term capturing the impact of 16 covariates of interest–data compiled from multiple sources (see Section 1 of the S1 Appendix) capturing: the population ethnicity proportions, the index of multiple deprivation (IMD) scores, the population age proportions, the population densities, the median annual incomes, the time spent at certain locations, the proportion of new COVID-19 variants, and the amount of COVID-19 funding allocated, across all 306 LTLAs considered. The impact of these variables is captured in the term $x_{i,t}\beta$, where $x_{i,t}$ represents the 16 covariates introduced above for LTLA $i$ at week $t$, and $\beta$ is a model parameter of coefficients controlling the relative contributions of each of the 16 covariates.

Fig 3 shows the mean and 95% CrI of the posterior distributions for these covariate coefficients (parameter $\beta$ above). Model covariates ($x_{i,t}$) were standardised to have mean 0 and standard deviation 1 before model fitting to enable comparison of relative covariate coefficients.

Unsurprisingly, the time spent at home was the strongest covariate effect (outside of COVID-19 variant) in determining changes to transmission. LTLAs and weeks where populations spent more time in residential areas saw reduced effective reproduction numbers. Similarly, LTLA-weeks with more visits to (non-home) workplaces saw increased reproduction numbers. Additionally, the LTLAs with greater allocations of Adult Social Care (ASC) infection control funding per head saw reduced reproduction numbers.

Each new variant was associated with a sequential increase in the reproduction number, in alignment with similar studies into the transmission potential of each variant [10].

Our analyses suggested there was no statistically significant impact on the effective reproduction number by population ethnicity proportions, IMD, population proportion over the age of 65, population density, median annual income, visits to transit stations, un-ringfenced funding or Contain Outbreak Management Fund (COMF) funding allocated–all these coefficients' 95% CrI overlaps 0 in Fig 3.

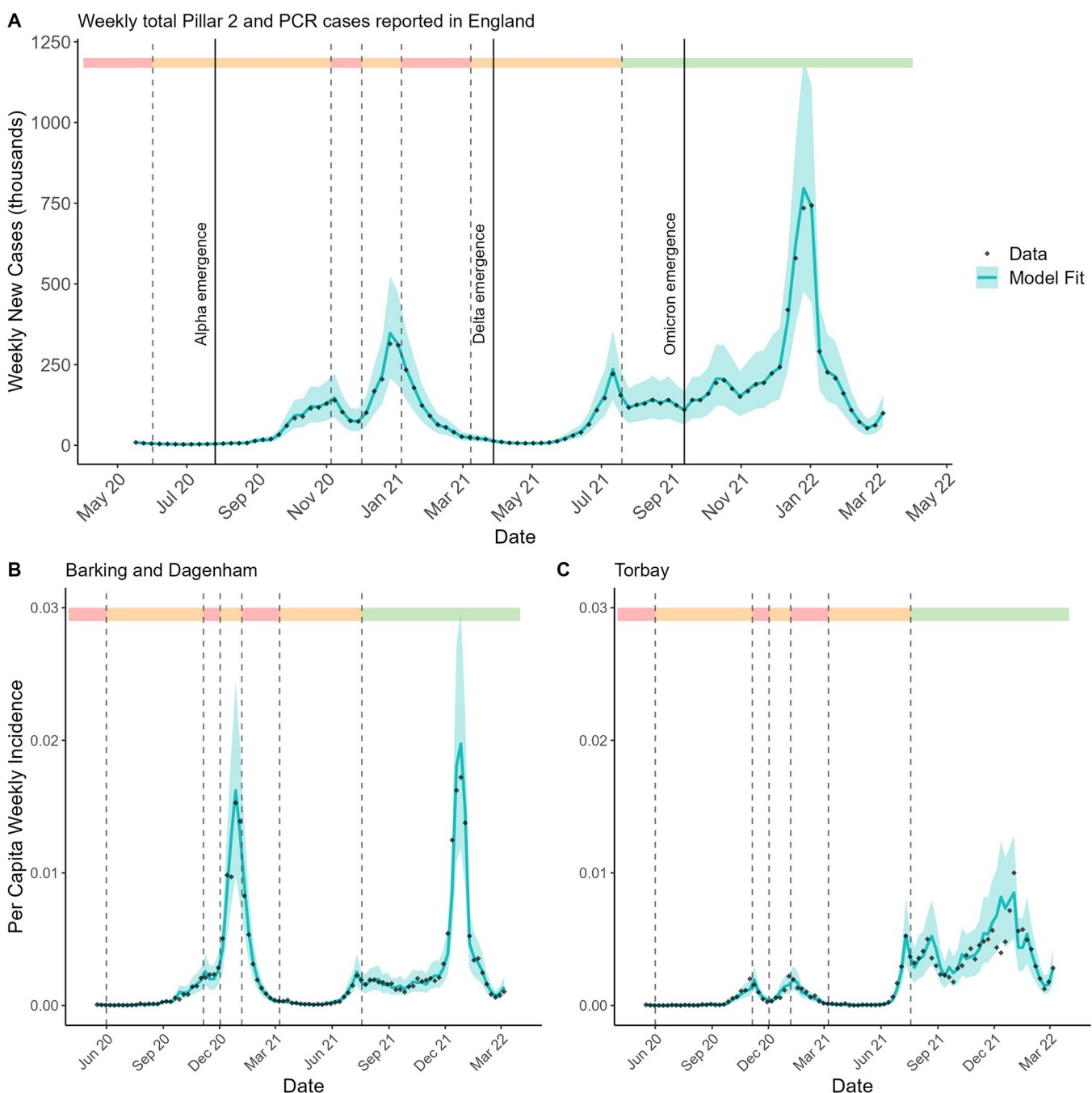

**Fig 2. Model fit to data, at national level and two LTLAs as examples. (A)** The number of weekly new COVID-19 cases in England, summing the model fit across all 306 LTLAs. **(B/C)** The per capita weekly incidence of COVID-19 in two specific LTLAs with greatly different epidemic trajectories. Black dots show data and the blue line represents mean model fit. Shaded blue regions depict the 95% credible intervals (CrI). Dashed lines represent significant changes in nationwide non-pharmaceutical interventions imposed [9]. As shaded in the top bar, red areas depict times of full "stay-at-home" orders, orange depicts partial restrictions on social mixing, green depicts no barriers to social mixing.

Our model also assumes that the number of weekly cases in an LTLA is not just driven by the previous weekly number of cases in that LTLA, but that some new infections can be triggered by infections in adjacent LTLAs. This is a process known as spatial exportation, whereby a primary case in one LTLA may visit a neighbouring LTLA, and subsequently cause a

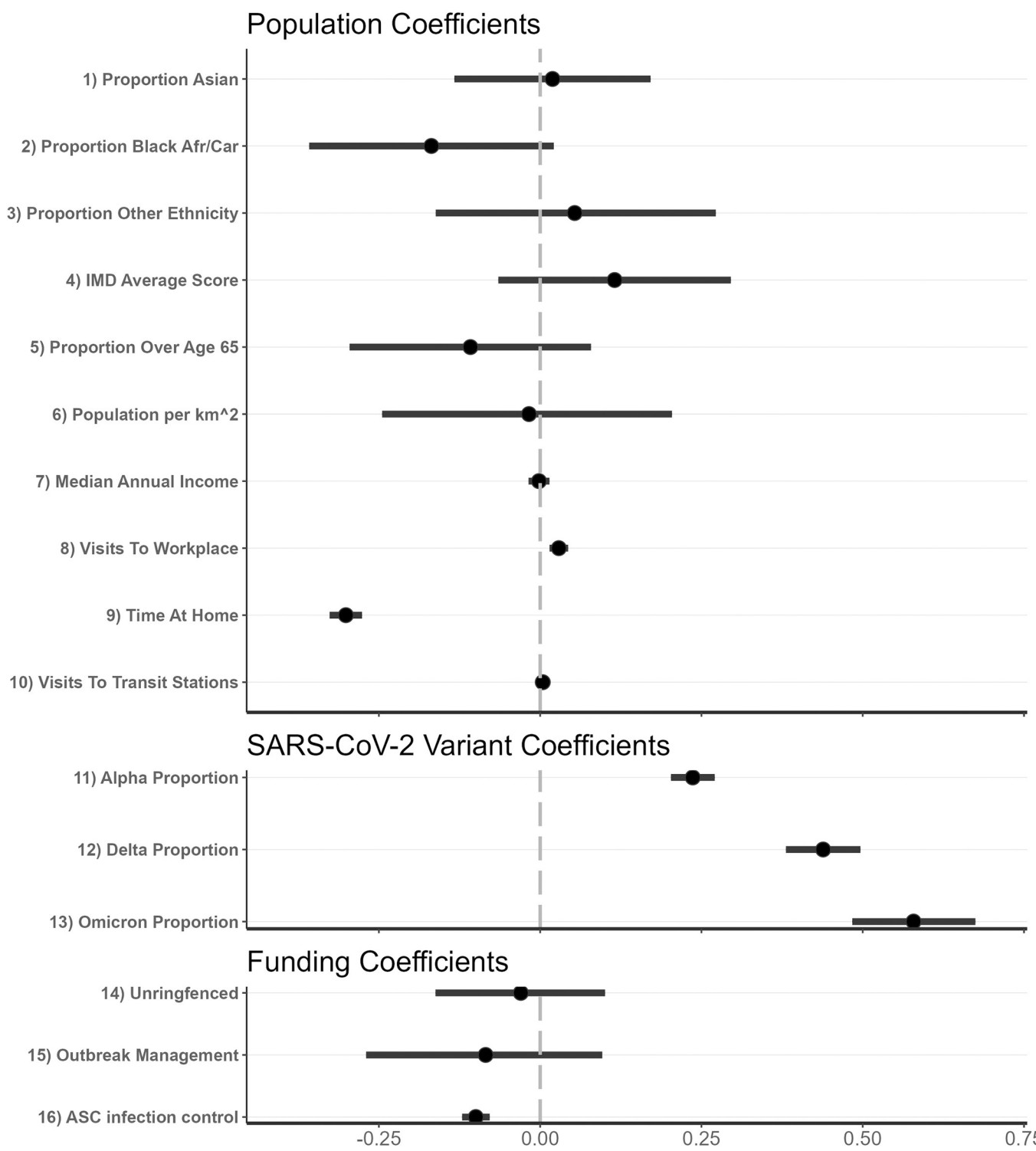

**Fig 3. Posterior estimates of all covariate coefficients (parameter $\beta$).** These values are separated into three categories: those capturing population effects, those capturing variants-of-concern, and those capturing funding allocations. Black dots represent the mean estimate, black lines the 95% CrI. The dashed grey line marks 0. A positive value indicates that the effective reproduction number increases with higher values of the associated covariate, a negative value indicates that the effective reproduction number decreases with higher values of the associated covariate.

secondary infection outside of their home boundaries. We assume that the proportion of spatial importations varies by LTLA, as some areas, like city centres, may attract more visitors than other, more rural, LTLAs. The model parameter $\zeta_i$ is defined as the proportion of all weekly cases in adjacent LTLAs that contribute secondary infections each week in LTLA $i$.

Fig 4 shows the impact of all other model variables that contribute to the effective reproduction number–the proportion of case importations from neighbouring LTLAs, $\zeta_i$, (median

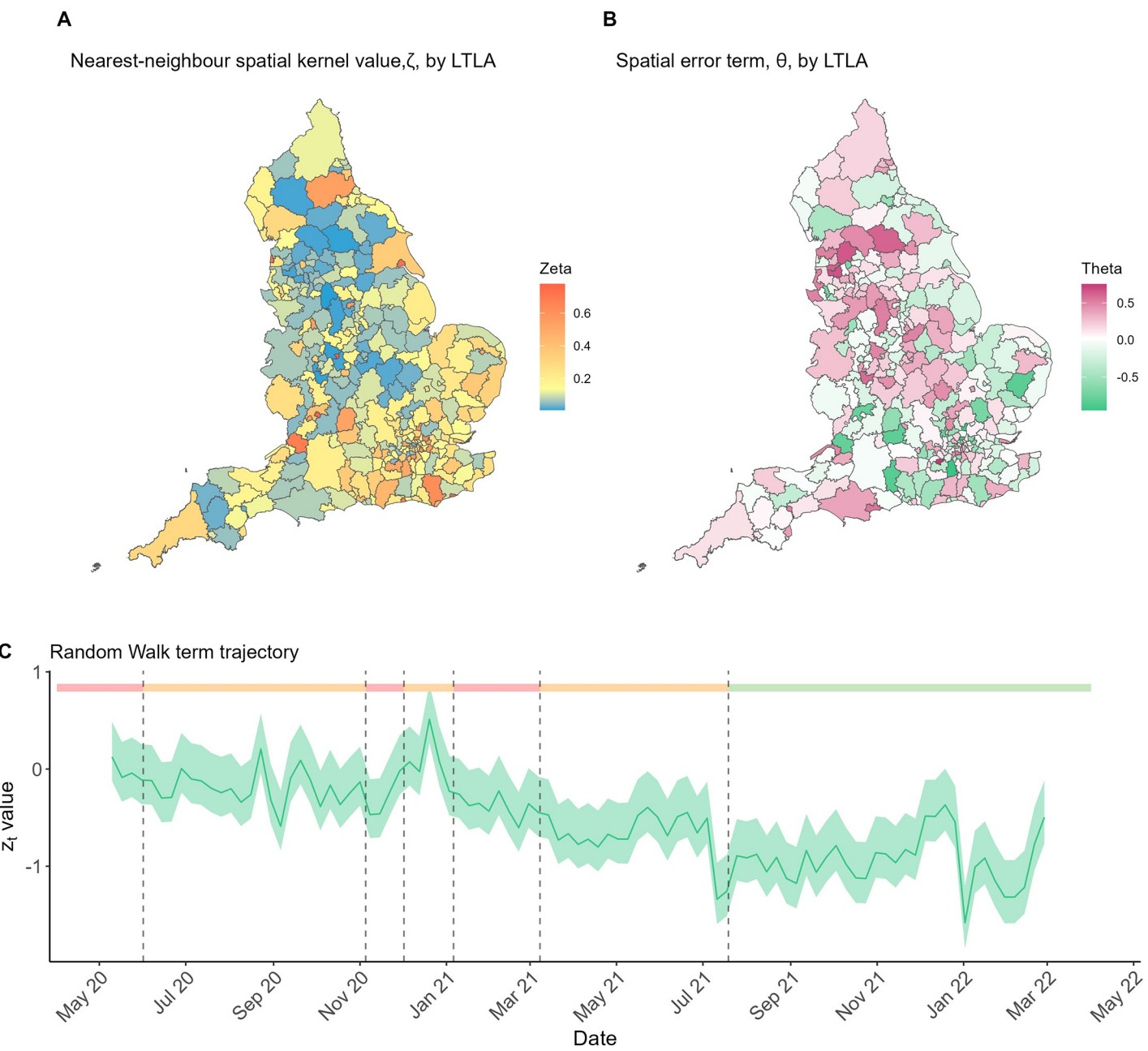

**Fig 4. Model variables contributing to the LTLA-varying and week-varying effective reproduction number.** (**A**) Estimates of the LTLA-varying parameter $\zeta_i$, denoting what proportion of the cases in adjacent LTLAs cause secondary cases in LTLA $i$ the following week. (**B**) The remaining spatial error term, $\theta_i$, capturing underlying differences in LTLA reproduction numbers not explained by the sixteen considered covariates ($x_{i,t-1}\,\beta$). (**C**) The random walk term applied to all LTLAs capturing a baseline time-varying change to the reproduction number. Solid line shows the mean estimate and the shaded region the 95% CrI. Dashed lines and shaded bar at the top of the plot again mark areas of full (red), partial (orange) and no (green) stay-at-home orders. Boundary source: Office for National Statistics licensed under the Open Government Licence v.3.0[29]. Contains OS data crown copyright and database right (2024).

value 0.134, interquartile range (IQR) 0.069–0.247) the LTLA-varying error terms $\theta_i$ (median value 0.028, IQR -0.159–0.222), and the time-varying random walk trajectory, $z_t$ (median value -0.489, IQR -0.871 - -0.243). Fig 4A shows that the majority of LTLAs imported only a small proportion of cases from neighbouring LTLAs—LTLAs shaded in blue saw less than 20% of the total cases in LTLAs they share a border with causing onwards infections within their own boundaries. Denser populated LTLAs like city centres have higher $\zeta_i$ values in general, (areas shaded red in Fig 4A) demonstrating the increased transmission risk caused by individuals travelling into population centres from more rural LTLAs. The greatest contribution of the three terms comprising our $R_{i,t}^{eff}$ is thus the random walk term, followed by the covariate impacts and LTLA-specific error term $\theta_i$.

Fig 4B shows the spatial variation in the reproduction number that is not explained by the 16 covariates ($x_{i,t-1}$). The apparent clustering of higher and lower $\theta_i$ values suggests this pattern may not simply be noise, but that there could be spatial variables yet to be identified that may also be influencing the transmission of COVID-19 across LTLAs in England.

Fig 4C shows the random walk trajectory over time. There are multiple time-varying aspects that will have influenced the epidemic in England. We specifically include COVID-19 variant proportions, and our mobility covariates have been shown to capture the influence of NPIs over time [11]. Fig 4C demonstrates a gradual decrease in the background effective reproduction number for all LTLAs, in line with the uptake in vaccination and changes in behaviour.

## Model comparison

To assess the relative contribution of each of these modelled mechanisms towards improving model fit, the model is fit multiple times with some mechanisms removed.

Alongside the main analysis (MA) presented above, where all model terms are included, three additional sensitivity analyses are presented: (SA1) we fit the model without any covariates included–i.e. $\beta$ is hard-coded to 0, (SA2) we fit the model with no spatial exportation included (but covariates are still included)–i.e. $\zeta_i$ is hard-coded to 0, (SA3) we fit the model with no spatial exportation or covariates included–i.e. both $\beta$ and $\zeta_i$ are fixed at 0. $\theta_i$ and $z_t$ remain in all three sensitivities.

Model comparison is performed using the expected log pointwise predictive density (elpd) metric under a "leave future out" (LFO) cross-validation scheme, detailed in section 4.2 of the S1 Appendix. elpd (LFO) assesses the relative goodness of fit and predictive performance of different model versions by evaluating each model's ability to predict held-out future sections of the time series. Higher elpd (LFO) values (lower magnitude) indicate better model performance.

The elpd (LFO) values for the main analysis (MA) and three sensitivity analyses (SA1-3) are shown in Table 1 below. A greater elpd (LFO) value suggests a better model fit.

The greatest elpd value, and hence the best performing model, is the main analysis (MA), containing both a nearest-neighbour spatial importation mechanism, and population / variant

**Table 1. Model comparison via the elpd leave-future-out cross-validation measure.** A greater value suggests a better model fit. "Spatial importation excluded" indicates that $\zeta_i$ is fixed to 0, and "covariates excluded" indicates that $\beta$ is fixed to 0. When included these are both fit model parameters.

| Model Formulation | elpd (LFO) value (point estimate / standard error) |
| --- | --- |
| (MA) Spatial importation included, covariates included. | -81,006 (SE 505) |
| (SA1) Spatial importation included, covariates excluded. | -81,124 (SE 502) |
| (SA2) Spatial importation excluded, covariates included. | -83,945 (SE 667) |
| (SA3) Spatial importation excluded, covariates excluded. | -84,018 (SE 666) |

/ funding covariates. This is to be expected as the sensitivities are nested models of the main analysis, and the elpd does not directly penalise increased model complexity. However, the improvement offered by including the model covariates (MA vs. SA1, and, SA2 vs. SA3) is insignificant once the standard errors in the elpd estimates are considered, and therefore not worth the complexity trade-off of their inclusion. The improvement offered by the inclusion of spatial importation mechanisms however (an estimated elpd increase of 2,939, MA vs. SA2) are significant, and support the inclusion of spatial importation as important to explaining variation in disease incidence between LTLAs.

Other model formulations are included as supplementary results for comparison in section 4 of the S1 Appendix, including alternate spatial kernels, alternate data sources, univariate models, and consideration of different reporting assumptions.

## Discussion

Our study explored the informative potential of multiple spatially varying health inequity, socio-demographic, and socio-economic factors on week-to-week transmission potential within a population. We investigated how these variables related to the observed differences in COVID-19 week-to-week transmission across 306 administrative areas of England over a period of 95 weeks. In conclusion, the majority of these variables were not found to be significantly associated with COVID-19 transmission; however, we did detect a significant association for two population variables–the time spent at home, and the number of visits to workplaces, and one funding variable–the amount of ASC infection control funding allocated per head to an LTLA.

Starting with ethnicity, Black and South Asian populations have been shown to have increased COVID-19 mortality risk [12]. In their global systematic review of the impact of ethnicity on COVID-19 health outcomes, Irizar et al. (2023) [13] report mixed results when comparing the risk of infection for Asian and "other" ethnicity populations with White majority populations, in accordance with our results in Fig 3. However, they show a far more conclusive increased relative risk in Black populations compared to white majority populations. Our results suggest a mild decrease in week-to-week transmission potential for LTLAs with a higher proportion of Black African / Caribbean residents on average (though still statistically insignificant). One possible explanation is that put forward by Harris & Brunsdon (2021) [14], who show that the distribution of COVID-19 cases by ethnicity changes over time in England, with Black populations reporting far higher relative incidence during the peak of the first wave, before then changing to capturing the minority of cases proportionally. Similarly, Mathur et al. (2021) [15] report a lower risk of infection in Black populations compared to White populations during the second wave. They demonstrate this is likely due to the heterogeneity in spatial incidence over time. Our study both directly factors for spatial heterogeneity in incidence and considers a longer time period than these studies investigating this association, potentially explaining our finding. Section 4.3 of the S1 Appendix shows that the mean coefficient magnitude for the population Black African / Caribbean proportion is further reduced in the absence of all other covariates, suggesting that some degree of covariate correlation is also influencing the estimated importance of the covariate.

Crucially; however, care must be taken when comparing our results to those of community-targeted infection risk studies. Population prevalence studies, such as the REal-time Assessment of Community Transmission (REACT) studies, and those conducted via the OpenSAFELY platform, directly investigate how COVID-19 prevalence differed by demographic indicators like those considered in this study. Ward et al. (2021) [16] identified a three-fold increase in testing antibody-positive within Black populations compared to White populations

(reducing to two-fold when adjusted for confounding factors such as age, sex, IMD quintile, household size). Mathur et al. (2021) [15] identified a similar risk for the period February 1st–August 3rd 2020, though this increased risk is not identified for the "second wave" of September 1st–December 31st 2020. Such results should not be directly compared to the findings of this study, which investigates a fundamentally different result–we do not consider denominator populations, or individual-level infection results; rather, we consider how the composition of a population contributes to week-to-week transmission potential.

Our results also show an inconclusive impact of IMD on transmission, though lean towards higher reproduction numbers seen in more deprived areas. In their systematic review of socio-economic COVID-19 impacts, Benita et al. (2022) [17] list only nine UK-specific studies, and report a global trend of mixed and inconclusive findings as to the impact of poverty metrics on COVID-19 infection. The trend we have shown in Fig 1B, of differences in case incidence by deprivation quantiles seeming pronounced in some time periods, before reversing in others is seen in multiple other countries [18,19].

The strong negative effect of the "time spent at home" variable is unsurprising given its inherent epidemiological importance, and its direct impact on disease cases has been demonstrated for multiple countries [20]. While this variable predominantly changes temporally in relation to NPI measures, strong variation is also observed across LTLAs at any given time point (see section 1.6 of S1 Appendix). While it is possible that the impact of some other covariates is captured within the "time spent at home" variable, i.e. LTLAs with higher incomes also see more time spent at home on average, additional sensitivity analyses exploring the removal of this variable in the S1 Appendix (section 4.3) shows that results are broadly unchanged by their inclusion.

Of the three COVID-19 funding pools provided during the pandemic that we consider, only the ASC infection control fund proved significant. COMF funds are provided for activities such as targeted testing of hard-to-reach bodies, additional contact tracing, community support, communication materials, as well as enforcement and compliance expenditure [21]. The ASC infection control fund meanwhile was specifically for use in preventing onward transmission in care home settings. As with many European countries, care homes in England were hit particularly hard during the first wave of the epidemic in England [22], motivating this specific fund. Our results show that this specific targeting of the most vulnerable populations was effective in reducing transmission and is the first study to our knowledge to investigate the associated impact of these funding provisions. All funding allocations were informed by the specific health needs and population demographics of respective LTLAs, meaning that some covariates such as IMD and age distribution will have some degree of correlation.

To investigate the time-specific impact of these covariates, we conducted a sensitivity analysis whereby the model was fit to three distinct subsections of the overall time series (section 4.5 of the S1 Appendix). Our results are unchanged across specific time periods, save for the significance of the ASC infection control fund disappearing for August 8th 2021 onwards, as would be expected, as this was when NPIs had been lifted.

Section 1.7 of the S1 Appendix presents the degree of autocorrelation present amongst variables. This unavoidable aspect of the dataset is a limitation of the study; however, we address this through multiple supplementary sensitivity analyses including univariate model formulations. All fundamental results presented in this study are maintained under these sensitivities, save for the impact of IMD, which does achieve statistical significance upon exclusion of the other fifteen covariates.

We have not included vaccination directly, as the vaccination rollout was itself influenced by the epidemic trajectory, with greater dose uptake encouraged in response to novel variants [23]. As such, since our model does not mechanistically include the effect of vaccination, nor

the impact of waning effectiveness, the random walk term will capture both the uptake and impact of vaccination, but also unique temporal aspects such as public holidays, sporting events, seasonal patterns, and others. We see in general a reduction in the reproduction number over time, in line with the vaccine rollout, but also note increases in December likely aligned with the Christmas holidays, and other adjustments such as an increase in June/July 2021 around the time of the UEFA European Football Championship [24], followed by a drop after the event.

A caveat of this study is the heterogeneity within each LTLA for some covariates of interest. LTLAs are areas of geographic administration and service provisioning, and as such differ in population sizes. While many LTLAs are close to the median LTLA population size of 142,622 people (IQR 104,869–237,616, see section 1.8 of the S1 Appendix), some outliers are considerably different, the largest being Birmingham with a population of 1,140,525. Heterogeneity in covariates, such as IMD, within these larger LTLAs can be observed at the Lower layer Super Output Areas (LSOAs) scale, of 32,844 areas in England, however COVID-19 case data at this scale is too sparse to model. Thus, the LTLA-scale considered demonstrates a trade-off between demographic detail, data availability, and modelling feasibility.

We also note a modelling assumption made whereby reported "first episodes" of disease incidence in an LTLA contributed to fully immunising protection against onwards infection. While protection against repeat infection was strong for the majority of the time period we considered [25], this was likely to wane more against the Omicron variant. We explored this modelling assumption through multiple sensitivity analyses where the model was fit to different time periods, and where waning of acquired immunity was assumed, which are presented in sections 4.5 and 4.6 of the S1 Appendix. Our model results were unchanged in these analyses.

The overarching question motivating this study was whether population health and demographic variables held informative potential such that their inclusion might improve real-time modelling efforts that currently do not incorporate such data streams. Only a minority of covariates were found to be impactful, and the improvement to model fit they offer is insignificant. However, substantive improvements are offered by including mechanisms of spatial spread. Detailed study of lineage exports by Kraemer et al. (2021) [26] have previously demonstrated how human travel alone was able to explain the spatial heterogeneity observed during the emergence of the Alpha variant in the UK, further supporting our findings. The notable impact of our simplified adjacency kernel would likely be strengthened further were it supplemented with such direct measures of individual mobility patterns.

It is important to note that the highlighted impact of epidemiological factors, such as mobility patterns and contact networks, does not exclude the importance of health inequity in pandemic response. Health inequity is a complex phenomenon that extends beyond aggregated measures of deprivation and access-to-care, and further work is required to investigate its relation, both socially and quantitatively, to these underlying epidemiological factors. Our results should not be interpreted as ruling out the impact of health inequity in shaping epidemic trajectories; rather, that the aggregated measures considered do not directly improve model prediction of week-to-week transmission within a geography.

While real time modelling efforts are often limited by computational power and thus are limited in what level of spatial disaggregation can be allowed for, we have effectively demonstrated that mechanisms of case exportation are a worthwhile inclusion for improving model fit, and that the benefits of incorporating broader socio-demographic data are unlikely to be worth the time needed to gather and incorporate the relevant and up-to-date data.

## Methods

### Ethics statement

Ethics permission was sought for the study via Imperial College London's (London, UK) standard ethical review processes and was approved by the College's Research Governance and Integrity Team (ICREC reference 21IC6945). Patient consent was not required as the research team accessed fully anonymised data only, which were collected as part of routine public health surveillance activities by the UK Government.

### Study population and data

Confirmed COVID-19 cases data were taken from the UK Health Security Agency (UKHSA) national line list, collected by the Department of Health and Social Care as part of surveillance activities and shared with us. Only pillar 2 cases (swab testing of the wider population, not setting-specific) confirmed via PCR were used to account for changes in the availability of lateral flow devices (LFDs), as well as changes in test-seeking behaviour. Cases were then aggregated by week (beginning Monday) and LTLA. S-gene target failure (SGTF) data for each case was similarly obtained from the UKHSA line list to identify the proportion of COVID-19 variants each week. The cumulative number of first episodes by LTLA is obtained from the national data dashboard. Population data on ethnicity, age, population density, income, was taken from Office for National Statistics (ONS) reports. IMD data is taken from the Ministry of Housing, Communities & Local Government (MHCLG) report on English Indices of Deprivation 2019 (IoD2019). Data on time spent at locations is taken from Google community mobility reports. Data on COVID-19 funding allocations was taken from the associated Department for Levelling Up, Housing and Communities reports.

In 2021 England was split into 309 LTLAs. Following the format used for the COVID-19 cases data release, we combine the LTLAs of Cornwall and Isles of Scilly; City of London and Hackney. The Isle of Wight LTLA is removed. Thus, this study reports on 306 English LTLAs total.

Detailed descriptions of all covariates are provided in section 1 of the S1 Appendix.

### Epidemiological model and fitting

Using Bayesian evidence synthesis inference we fit a probabilistic model to data $Y_{i,t}$, the number of weekly pillar 2 PCR-confirmed COVID-19 cases in LTLA $i$ at week $t$, via a negative binomial distribution of the form

$$Y_{i,t} \sim NegBinom(\mu_{i,t}, \phi)$$

for mean $\mu_{i,t}$ and overdispersion parameter $\phi$. The mean takes the form

$$\mu_{i,t} = \left( \lambda S_{i,t-1} \left( Y_{i,t-1} + \zeta_i \sum_{j \in \Omega_i} Y_{j,t-1} \right) \exp(x_{i,t-1}\beta + z_{t-1} + \theta_i) \right)$$

where $S_{i,t-1}$ is an estimate of the proportion of the population of LTLA $i$ that has no acquired immunity in week $t-1$, calculated as 1 –(total number of recorded "first episodes" in LTLA $i$ by week $t$ / LTLA $i$ population), and $\lambda$ is a scaling factor parameter, between 0 and 1, scaling the acquired-immunity lag term to account for the impact of under-reporting of first episodes, incomplete protection of acquired immunity, and other nationwide scaling effects. $\Omega_i$ is the set of all LTLAs that share a boundary with LTLA $i$, and $\zeta_i$ is a model parameter between 0 and 1 denoting the proportion of cases in neighbouring LTLAs which will cause secondary cases in LTLA $i$. This represents a "nearest neighbours" spatial kernel formulation. $x_{i,t-1}$ is a vector of

the sixteen covariates considered in this study for LTLA $i$, at week $t−1$. $\beta$ is the vector of coefficients capturing the relative impact of each covariate. A covariate is considered statistically significant if the 95% CI of its $\beta$ coefficient posterior distribution does not span 0. $z_{t−1}$ represents the $(t−1)^{th}$ step in a Gaussian random walk process, and $\theta_i$ is an LTLA-specific error-term.

Heuristically, the left-hand side of the expression represents the number of cases contributing to the next week's number of cases, and the right-hand side may be considered an estimate of the time-varying reproduction number.

We model 95 weeks in total, from the week beginning May $10^{th}$ 2020 to the week beginning February $27^{th}$ 2022, as case testing rates become inconsistent outside of this window.

Analyses were conducted in R version 4.1.1. The model was run in Stan via the rstan [27] package. All associated code is available in our online repository (https://github.com/thomrawson/Rawson-spatial-covid). See section 5 of the S1 Appendix for full details of package versions.

Further methodological detail is provided in sections 2 and 3 of the S1 Appendix.

### Sensitivity analyses

Model comparison is performed via the expected log pointwise predictive density (elpd) score under a leave-future-out cross-validation process, detailed in section 4.2 of the S1 Appendix.

As a supplementary result we also test the impact of using different data streams. While SGTF is considered a highly accurate indicator for discerning between variants of concern (VOC) [28], for completeness the S1 Appendix also presents analyses where variant proportion is instead confirmed by whole genome sequencing (WGS). We also conduct a sensitivity analysis where case data is expanded to include both pillar 1 and pillar 2 cases and includes LFD cases. In both sensitivities, the results remain unchanged from the inclusion of these data.

Sensitivity analyses exploring acquired-immunity assumptions are presented in section 4.6 of the S1 Appendix.

Sensitivity analyses exploring differences in reporting by ethnicity and IMD are presented in section 4.7 of the S1 Appendix.

Other model formulations are included for comparison in section 4 of the S1 Appendix, including alternate spatial kernels and univariate models.

## Supporting information

**S1 Appendix. Supplementary Material.**
(PDF)

## Acknowledgments

We thank all colleagues at the UK Health Security Agency (UKHSA, formerly Public Health England) and front-line health professionals who have not only driven and continue to drive the daily response to the COVID-19 epidemic in England but also provided the necessary data to inform this study. This work would not have been possible without the dedication and expertise of said colleagues and professionals. The use of pillar-2 PCR testing data and the variant and mutation data was made possible thanks to UKHSA colleagues.

For the purpose of open access, the author has applied a 'Creative Commons Attribution' (CC BY) licence to any Author Accepted Manuscript version arising from this submission.

## Author Contributions

**Conceptualization:** Thomas Rawson, Neil M Ferguson.

**Data curation:** Thomas Rawson, Wes Hinsley.

**Formal analysis:** Thomas Rawson.

**Investigation:** Thomas Rawson, Neil M Ferguson.

**Methodology:** Thomas Rawson, Raphael Sonabend, Elizaveta Semenova.

**Resources:** Thomas Rawson, Wes Hinsley.

**Software:** Thomas Rawson, Wes Hinsley, Raphael Sonabend, Elizaveta Semenova.

**Supervision:** Anne Cori, Neil M Ferguson.

**Visualization:** Thomas Rawson.

**Writing – original draft:** Thomas Rawson.

**Writing – review & editing:** Thomas Rawson, Wes Hinsley, Raphael Sonabend, Elizaveta Semenova, Anne Cori, Neil M Ferguson.

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
