## [Decision Letter · Decision Letter 0]

20 Feb 2024

Dear Dr Rawson,

Thank you very much for submitting your manuscript "The impact of health inequity on regional variation of COVID-19 infection in England" for consideration at PLOS Computational Biology.

As with all papers reviewed by the journal, your manuscript was reviewed by members of the editorial board and by several independent reviewers. In light of the reviews (below this email), we would like to invite the resubmission of a significantly-revised version that takes into account the reviewers' comments.

I agree with the reviewers that this is a potentially important contribution that likely advances our understanding of health inequity on both how epidemics unfold and how it alters our perception of how they unfold. However, I also agree that a significant amount of work is needed to revise the manuscript. The authors should focus on strengthening their model comparisons, the potential bias associated with using detected cases (e.g., could the authors use the serosurveys run regularly in the England during the pandemic), and a detailed analysis of whether their information theoretic approaches to model selection are justified. The reviewers provide a detailed set of thoughtful comments, which the authors should pay careful attention to during their revision.

We cannot make any decision about publication until we have seen the revised manuscript and your response to the reviewers' comments. Your revised manuscript is also likely to be sent to reviewers for further evaluation.

Sincerely,

Samuel V. Scarpino

Academic Editor

PLOS Computational Biology

Thomas Leitner

Section Editor

PLOS Computational Biology

I agree with the reviewers that this is a potentially important contribution that likely advances our understanding of health inequity on both how epidemics unfold and how it alters our perception of how they unfold. However, I also agree that a significant amount of work is needed to revise the manuscript. The authors should focus on strengthening their model comparisons, the potential bias associated with using detected cases (e.g., could the authors use the serosurveys run regularly in the England during the pandemic), and a detailed analysis of whether their information theoretic approaches to model selection are justified. The reviewers provide a detailed set of thoughtful comments, which the authors should pay careful attention to during their revision.

Reviewer's Responses to Questions

**Comments to the Authors:**

Reviewer #1: In the proposed manuscript, the authors use the COVID-19 linelist data from the UK to model the SARS-CoV-2 epidemic between 2020 and 2022 at the LTLA level. They assess the importance of including LTLA-specific covariates in explaining the epidemic, concluding that ultimately, there was a limited role of factors linked to deprivation, ethnicity in driving the epidemic, while the spatial process was much more important. Overall, this represents an interesting analysis applied to very detailed series of datasets that capture the complex factors linked to SARS-CoV-2 spread. I have a few comments.

Ultimately this remains an ecological analysis comparing crude summary statistics from an area with incidence of detected cases from that area. In particular, deprivation is based on quantile – that is summarised for an area with over 100k people. There will certainly be heterogeneity within each LTLA that gets lost at this level of resolution. Community-based infection studies, such as the ONS and REACT studies that directly measured underlying levels of infection, provide a more direct link to individual infection risk. These community studies have tended to show a big difference in infection risk as a function of ethnicity, social deprivation, household size etc, especially early in the epidemic (see e.g., Ward et al., Nature Communications 2020). You can certainly get different conclusions when exploring the overall population level effect and individual infection risk – however, the way the paper is written, this nuance is lost. I would be careful to ensure that readers don’t conclude that there was equal infection risk by ethnicity, social deprivation, especially early in the epidemic, as this certainly wasn’t the case as shown by these direct measures. I would also include a discussion of the comparison between inferences from these community-based infection studies and this study.

The beta vector that links the covariates is static in time – however, some of these covariates could have changed importance over time. This should be discussed.

The susceptibility approach makes me nervous. The model assumes an immunizing infection and that the impact of vaccination is absorbed in the zi term. However, the zi term is the other side of the equation to susceptibility. We also know that reinfection was common, especially across variants. The authors demonstrate a big effect of variant proportion on case numbers – but this is difficult to interpret in the context of assumed sterilizing immunity between variants.

Is there an assumption of equal reporting by covariate? So if individuals of a lower deprivation systematically reported less, would that be captured in the LTLA specific term?

It is important to note that the majority of infections were probably missed by LTLA. This study focuses just on detected cases – however many infections were subclinical or not detected via PCR (and instead by at-home antigen testing) – especially towards the end of the time series. I suspect this all gets absorbed in the zi term, however, it would be good to at least mention the role of undetected infections.

Line 40. It would be good to get a sense of the size of LTLA (e.g., mean number of inhabitants). Is it reasonable to assume the population is homogeneous by the covariates of interest within an LTLA?

Line 67. The authors use an adjacent spatial model. I can see the reasons for doing this – however, is this the most relevant if we consider that e.g., parts of Manchester may be more connected to London than neighbouring LTLAs, and that this connectivity may change over the course of the pandemic?

Line 73. It is worth noting that real-time modelling may have different roles at different stages of an epidemic – especially one lasting several years. The presented results are for an overall average – however, early in the epidemic, when there are many more uncertainties, the model result may look different.

Figure 1B/1D – are these consistent with what was inferred from the community infection studies (ONS/REACT)? Is there a strong assumption about equal testing probability across the sub-populations and over time behind these plots?

Figure 3. I wonder again the importance of these terms at different stages of the epidemic. If the purpose of the paper is to explore a role for modelling in outbreak response, the authors may wish to present how these estimates change (if at all) at different stages of the pandemic – rather than an overall average based on data up to a time by when pretty much everyone in the country had been infected, often multiple times.

Figure 4. This is an interesting figure. I believe it is showing that the majority of the variance is captured by these ‘catch all’ temporal and spatial terms. Would it be possible to quantify how much of the overall variance in incidence is captured by each term? It may be that there is substantial heterogeneity at smaller spatial scales, time varying coefficients that are needed. Again – if the purpose is to understand the role of real time modelling – can these parameters be estimated reliably in an outbreak setting and what do they ultimately say?

Line 182- slightly pedantic point but diseases don’t spread - pathogens spread.

Line 183 – is Zt correlated with vaccine uptake (it seems this data is available)? – I assume Zt covers all manner of factors driving the epidemic.

Line 232 – I’d be very careful with this statement. It is important to note that the study is modelling detected cases, not infections. If there are systematic differences by ethnicity in reporting, systematic differences in symptom risk, this would impact these estimates. In the reference cited by the authors – the paper only uses outcome measures (mortality) without considering underlying differences in the risk of infection by ethnicity – which we know was significant in the early stages of the epidemic. For example, Ward et al., Nature communications 2021 used the REACT study results to show very limited difference in mortality risk once underlying differences in infection risk were accounted for.

Reviewer #2: (This review is also attached as a PDF which includes full formatting and math typesetting)

# Summary

The authors construct a probabilistic model that accounts for time-varying incidence across regions of England. Their model accounts for area-level risks as well as spatial connectivity between regions. They find that spatial connectivity or exportation was the primary factor driving spatial variation in risk. Shifts in dominant variant and other high-level factors (e.g. vaccine availability) were found to be key drivers of variation in incidence over time. A key goal of the study is to determine whether efforts at real-time or near real-time modeling could be improved by deeper integration of regional sociodemographic data into similar models.

Overall, this is a well thought-out, exhaustive, and technically sophisticated analysis. The authors should be commended for the thoughtfulness and thoroughness of their work. My comments and concerns outlined below should be interpreted in their intended spirit, which is to push the authors to sharpen the explanation of their results by making a more detailed comparison between their very flexible models including spatially structured and unstructured random effect terms and those relying more on locally-measured covariates. My primary concern relates to their reliance on an approach to model selection which may be inappropriate given violations of assumptions of exchangeability important to the validity of the LOO-PSIS framework, as well as conceptual difficulty in making use of their model comparison results to choose among the presented models. I will explain all of this in detail below.

# Major Comments

**Model Structure** The parameter $\\lambda$ is introduced as representing the degree of undercount in the number of first infections. I think this is likely too concrete an interpretation of this parameter, as it could also be adjusting the impact of prior infections on the number of subsequent infections in a given area due to uneven or incomplete protective immunity. In all likelihood, this parameter can be best understood as a kind 'fudge factor' that is scaling the immunological lag term represented by $S_{t-1}$ and includes some combination of the effect of prior infection on future immunity and the rate of undercounting. The authors should clarify the interpretation and role of this parameter in the model.

**Robustness of findings regarding local social factors.** One of the more unexpected findings of this study is that the LTLA-specific sociodemographic covariates did not have a strong association with area-level variability in risk. The authors conduct extensive sensitivity analysis to test the robustness of this finding to different model specifications. As evidenced by Table S4 in the supplement, the goodness of fit as measured by LOOIC is worst for models excluding the spatially-varying lag term ($\\zeta_i$). The differences are less dramatic for models including either a spatially varying or stationary spatial lag term ($\\zeta$).

While the authors helpfully include a sensitivity analysis showing the estimates of various social risk factors when controlled for individually vs. when all are simultaneously included in the model (Figure S60), this is only done in the presence of the spatially-varying importation term ($\\zeta_i$) and LTLA-specific random intercept ($\\theta_i$). Looking at Table S4, Model C, which includes spatially-varying $\\zeta_i$ but no area-specific $\\theta_i$ has a similar LOOIC value to Model A, which is the one presented in the most detail in the model. While the comparison in Figure S60 is helpful for illustrating that the correlations between these parameters do not impact their point estimates *in the presence of the spatial random effects*, the lack of detail on less-flexible models such as Model C or Models G & K makes it a bit difficult to fully assess their findings. Looking at the lower LOOIC values for models including the spatially unstructured random intercept $\\theta_i$ without covariates $x_i$ suggests that these flexible random effects are doing a quite good job of "mopping up" spatial variance and that the overall better LOOIC for the spatially flexible model may be an artifact of the choice to use the Pareto smoothed importance sampling (PSIS) along with leave-one-out cross-validation in which the information criterion is assessed one location/time combination at a time.

**Chosen information criterion may be inappropriate given model assumptions and inputs.** The authors state that they use the PSIS approach to computing LOO because the classic approach to LOO cross-validation is computationally expensive (i.e. leaving out one point at a time and re-running the model while computing the posterior predictive density for the held-out point). While this is true, I think it may be somewhat beside the point. The choice of pointwise LOO is unlikely to be appropriate in this setting for two reasons (for more detail, please see this document: https://mc-stan.org/loo/articles/online-only/faq.html#spatial) . First, the use of a spatial lag on the *observed* values of $y_i$ rather than the predicted latent values, as is present in the current model, violates the assumption of conditional independence of the observations $y_i$ necessary for LOO-based (with or without PSIS) to be valid in this setting. For spatial and time-series models (or those which are both as in this one), LOO is valid when the posterior predictive density is conditional on some latent value $f_i$ in which the likelihood $Pr(y_i |f_i, \\phi )$ is a function of the latent values (i.e. the $\\mu_i$ in the current model) and parameters related to the observation process ($\\phi$). How

---

## [Decision Letter · Decision Letter 1]

1 May 2024

Dear Dr Rawson,

Thank you very much for submitting your manuscript "The impact of health inequity on regional variation of COVID-19 transmission in England" for consideration at PLOS Computational Biology. As with all papers reviewed by the journal, your manuscript was reviewed by members of the editorial board and by several independent reviewers. The reviewers appreciated the attention to an important topic. Based on the reviews, we are likely to accept this manuscript for publication, providing that you modify the manuscript according to the review recommendations.

Sincerely,

Samuel V. Scarpino

Academic Editor

PLOS Computational Biology

Thomas Leitner

Section Editor

PLOS Computational Biology

Reviewer's Responses to Questions

**Comments to the Authors:**

Reviewer #1: The authors have done an excellent job of responding to my concerns. I have no further comments.

Reviewer #3: Reviewer 1 brought up several crucial points, many of which resonated with my own observations. I was impressed to see the revisions and the authors' thoroughness in conducting additional sensitivity analyses in great detail.

I have a couple of minor comments, along with one concern regarding the clarity of the results' interpretation (point 5):

1. How is statistical significance of variables defined?

2. Administrative regions or areas for LTLAs- needs consistency in the manuscript, including the title. https://www.ons.gov.uk/methodology/geography/ukgeographies/administrativegeography/england

3. The authors mention mechanistic impact in the abstract. How is this defined? I didn’t see it anywhere in the main manuscript or the supplementary methods.

4. In addition to Reviewer 1’s comment and the authors’ response to it “Line 67. The authors use an adjacent spatial model. I can see the reasons for doing this – however, is this the most relevant if we consider that e.g., parts of Manchester may be more connected to London than neighbouring LTLAs, and that this connectivity may change over the course of the pandemic?” I was hoping that the authors would consider mobility patterns to quantify connectivity, rather than Euclidean distances and a gravity model. I think it would be a stronger predictor if such a variable were to be used in real time. Or making a note of this in the discussion.

5. The last sentence in the abstract says ”While these results confirm the impact of some, but not all, measures of regional inequity in England, our work corroborates the finding that observed differences in regional disease transmission during the pandemic were predominantly driven by underlying epidemiological factors rather than the demography and health inequity between regions.” I think it is important to think about why the important drivers i.e. epidemiological parameters, mobility patterns could be different between areas. I would argue that inequity is a complex phenomenon (both socially and as a quantitative measure) and is probably a crucial underlying cause of the differences in epidemiological factors. This phrasing makes me nervous about its interpretation by readers. I would request the authors to be more specific in their concluding statement in the abstract which can be easily misinterpreted. It would be more appropriate and transparent to say that the static, aggregated sociodemographic variables don’t add much value in prediction of transmission or effective reproduction number. And this work would motivate the field to move forward in a direction where we try to understand the underlying causes and drivers of the epidemiological factors which are in turn the drivers of transmission. But from the point of view of prediction in real time and improving model fit (as described in the last section of the discussion), I agree that these poorly collected, outdated and often incomplete measures of inequity don’t add much value, as they currently stand. And I would be more confident with a similar careful phrasing in the abstract.

6. Is it possible that the random walk term captures many of the factors that constitute inequity and as expected is one of the greatest contributors to Reff?

7. The authors have made the distinction between detected cases and infections and provided sensitivity analysis for a few different reporting scenarios. Reporting (for lateral flow tests: sensitivity analysis 4.4 was done to include data from UK COVID-19 data dashboard) and test seeking behaviour (PCR which is the primary data used) likely changed over time and by the different groups. In addition to the analyses, it would be good make it clear that the Pillar 2 PCR dataset is likely a combination of the true (primarily symptomatic) cases, test seeking behaviour and changes in intended use of the tests (for symptomatic testing, confirmatory tests)– around line 78 or 287.

**Have the authors made all data and (if applicable) computational code underlying the findings in their manuscript fully available?**

Reviewer #1: Yes

Reviewer #3: **No: **The line list data was available to the authors through an agreement and has been noted in the Methods section, but all the code and publicly accessible data is made available.

PLOS authors have the option to publish the peer review history of their article (what does this mean?). If published, this will include your full peer review and any attached files.

Reviewer #1: No

Reviewer #3: No

Figure Files:

Data Requirements:

Reproducibility:

References:

---

## [Editor Report · Decision Letter 2]

7 May 2024

Dear Dr Rawson,

We are pleased to inform you that your manuscript 'The impact of health inequity on spatial variation of COVID-19 transmission in England' has been provisionally accepted for publication in PLOS Computational Biology.

Best regards,

Samuel V. Scarpino

Academic Editor

PLOS Computational Biology

Thomas Leitner

Section Editor

PLOS Computational Biology

---

## [Editor Report · Acceptance letter]

16 May 2024

PCOMPBIOL-D-23-01964R2 

The impact of health inequity on spatial variation of COVID-19 transmission in England

Dear Dr Rawson,

I am pleased to inform you that your manuscript has been formally accepted for publication in PLOS Computational Biology. Your manuscript is now with our production department and you will be notified of the publication date in due course.

With kind regards,

Anita Estes
